# Multimodal MRI Assessment of Thalamic Structural Changes in Earthquake Survivors

**DOI:** 10.3390/diagnostics11010070

**Published:** 2021-01-04

**Authors:** Federico Bruno, Alessandra Splendiani, Emanuele Tommasino, Massimiliano Conson, Mario Quarantelli, Gennaro Saporito, Antonio Carolei, Simona Sacco, Ernesto Di Cesare, Antonio Barile, Carlo Masciocchi, Francesca Pistoia

**Affiliations:** 1Department of Biotechnology and Applied Clinical Sciences, University of L’Aquila, 67100 L’Aquila, Italy; alessandra.splendiani@univaq.it (A.S.); emanuele.tommasino@gmail.com (E.T.); gennsaporito@gmail.com (G.S.); antonio.carolei@univaq.it (A.C.); simona.sacco@univaq.it (S.S.); ernesto.dicesare@univaq.it (E.D.C.); antonio.barile@univaq.it (A.B.); carlo.masciocchi@univaq.it (C.M.); francesca.pistoia@univaq.it (F.P.); 2Laboratory of Developmental Neuropsychology, Department of Psychology, University of Campania Luigi Vanvitelli, 81100 Caserta, Italy; massimiliano.conson@unicampania.it; 3Institute of Biostructure and Bioimaging, National Research Council, 80100 Naples, Italy; quarante@unina.it

**Keywords:** thalamus, trauma, stress, earthquake, MRI, DTI, DWI, brain volumes

## Abstract

Moving from the central role of the thalamus in the integration of inner and external stimuli and in the implementation of a stress-related response, the objective of the present study was to investigate the presence of any MRI structural and volumetric changes of the thalamic structures in earthquake witnesses. Forty-one subjects were included, namely 18 university students belonging to the experimental earthquake-exposed group (8 males and 10 females, mean age 24.5 ± 1.8 years) and a control group of 23 students not living in any earthquake-affected areas at the time of the earthquake (14 males and 9 females, mean age 23.7 ± 2.0 years). Instrumental MRI evaluation was performed using a 3-Tesla scanner, by acquiring a three-dimensional fast spoiled gradient-echo (FSPGR) sequence for volumetric analysis and an EPI (echoplanar imaging) sequence to extract fractional anisotropy (FA) and apparent diffusion coefficient (ADC) values. As compared to the control one, the experimental group showed significantly lower gray matter volume in the mediodorsal nucleus of the left thalamus (*p* < 0.001). The dominant hemisphere thalamus in the experimental group showed higher mean ADC values and lower mean FA values as compared to the control group.

## 1. Introduction

Thalamus works as an integrative center for multimodal stimuli, both of inner and external origin; it is bidirectionally linked with the prefrontal cortex and it also receives different inputs from subcortical structures like the striatum, the basal forebrain, the amygdala, and the brainstem, thus representing an essential multisensory relay for goal-directed behavior and emotional processing [1,2]. In particular, animal studies converge in demonstrating that specific thalamic regions, as the mediodorsal thalamus, relay information from subcortical visual structures, also including the superior colliculus, forming a tight loop with the prefrontal cortex and the amygdala [3,4,5] playing pivotal roles in the neurobiological mechanisms driving behavioral responses to threats and stressful events [6]. Indeed, the involvement of the thalamus in the stress system has been widely recognized. The stress system plays a crucial role in detecting threatening stimuli and in triggering the so-called fight-or-flight response, which is the prerequisite for survival. To serve this function, this system integrates the activity of the cortico–subcortical emotional pathways [2], resulting in the final activation of the sympathetic–adreno–medullar (SAM) axis and the hypothalamus–pituitary–adrenal (HPA) axis.

Exposure to natural disasters is undoubtedly a strong psychological stressor, which puts the involved subjects at risk of developing maladaptive cognitive and emotional changes [7]. Indeed, after a natural disaster, like an earthquake, people can develop different emotional disorders, the most frequently reported being post-traumatic stress disorder (PTSD), depression, and anxiety [8,9,10]. However, a certain percentage of individuals can develop behavioral and emotional changes representing subclinical pathological conditions [9,11,12]. Recently, Pistoia et al. [13] described a selective enhancement in the ability to read emotional facial expressions in L’Aquila 2009 earthquake survivors, suggesting hypervigilance to stimuli signaling a threat.

Magnetic resonance imaging (MRI) is undoubtedly one of the most useful tools to obtain information on morphological and functional aspects of the brain. Extensive research has shown the validity of blood-oxygen-level-dependent signals derived by fMRI studies to assess brain responsiveness to emotional stimuli [13]. Moreover, data from several voxel-based morphometry (VBM) studies revealed evidence of structural brain abnormalities associated with trauma [14,15,16].

Diffusion-weighted imaging (DWI) and diffusion tensor imaging (DTI) are established MRI sequences sensitive to the motion of water molecules within a voxel of tissue, and capable of estimating tissue cellularity and organization [17,18,19,20]. These techniques are extensively applied in routine clinical practice for the characterization and the evaluation of different brain and spine disorders such as tumors, ischemia, and white matter pathology [19,21,22].

Moving from the central role of the thalamus in integrating communication between cortical and subcortical structures, processing threatening visual stimuli, and implementing stress-related responses [1,2], the purpose of the present study was to investigate the presence of MRI structural changes of the thalamic structures, assessed through DTI, DWI, and VBM, in a cohort of earthquake survivors.

## 2. Materials and Methods

### 2.1. Participants

From the original sample of the previous neuropsychological pilot study [11], forty-one subjects (38% of the original sample) underwent MRI examination and were included in the present study. The subsample was composed as follows: 18 students belonging to the experimental earthquake-exposed group with a permanent residence in L’Aquila at the time of the 2009 earthquake (8 males and ten females, mean age 24.5 ± 1.8 years) and 23 students belonging to the control group not living in any earthquake-affected areas (14 males and nine females, mean age 23.7 ± 2.1 years). Non-parametric between-group comparisons showed that there were no differences in the mean age between the two groups. All participants in both groups were Caucasian Italian students.

Students were included according to the following inclusion criteria: (i) no history of previous or coexistent neurological or psychiatric diseases, including PTSD, as revealed by a psychiatric examination; (ii) no assumption of drugs or substances acting on the central nervous system; and (iii) signed informed consent to participate in the study.

### 2.2. MR Data Acquisition

MRI studies were carried out at three Tesla (Discovery MR, General Electric Medical Systems, Erlangen, Germany) using a 32-channel head coil.

Structural T1w volumes were acquired using a three-dimensional magnetization-prepared fast spoiled gradient echo sequence (FSPGR BRAVO) with the following acquisition parameters: 176 slices; TR 6.6 ms; TE 2.3 ms; TI 1100 ms; flip angle 7°; voxel size 1 mm^3^; matrix 256 × 256; FOV 25.6, duration 4:05 min. For DTI/DWI evaluation, an axial sequence (68 slices, slice thickness 2 mm, TR 15175, TE 100, matrix 128 × 128, FOV 24, duration 16:57 min) was acquired. In addition, turbo-spin-echo FLAIR axial images were acquired (144 sagittal partitions; TR 8000 ms; TE 119 ms; TI 2032 ms; flip angle 90°; voxel size 1 × 1 × 1 mm^3^) to rule out the presence of CNS pathologies.

### 2.3. VBM Analysis

Structural T1w volumes were manually oriented and automatically segmented in white matter (WM), grey matter (GM), and cerebral spinal fluid (CSF) using the segmentation tool implemented in Statistical Parametric Mapping software (SPM 12) running under MATLAB version 2018a. Default parameters incorporated in the DARTEL toolbox were used. We used the most used algorithm in SPM12 involving a 12-parameter affine transformation followed by a nonlinear registration using a mean squared difference matching function [23]. Gray matter images were then spatially smoothed with an 8 mm FWHM isotropic Gaussian kernel. Total intracranial volume (TIV), defined as the volume within the cranium including gray matter, white matter, and cerebrospinal fluid, was calculated to be used as a covariate, using the implemented tool in SPM12 (Figure 1).

### 2.4. FA and ADC Analysis

DTI and DWI images were preliminarily checked for excessive background noise, and echoplanar imaging (EPI) distortion correction was applied on a dedicated workstation. A processing threshold of 0.0083 was applied on the fractional anisotropy (FA) and apparent diffusion coefficient (ADC) map for the analyses of the thalami. All DTI/DWI images were co-registered to the B0 images, with gradient directions corrected for the applied directions. In both sides, the thalamus 3D extent was manually outlined on baseline DTI/DTI images in conjunction with high-resolution T1 images by two expert neuroradiologists (A.S., F.B.) blinded to demographic, clinical, brain dominance, and MRI data. Regions of interest were then placed and superimposed on the FA and ADC maps to extract the mean FA and ADC values for both thalami in each patient (Figure 2). Mean FA and ADC at baseline of each region of interest, left and right thalamus, were then indexed to the mean FA and ADC values of the thalamus according to the brain dominance.

### 2.5. Statistical Analysis

For the VBM analysis, the smoothed grey matter images were normalized to a global mean pixel-value of 50 and entered into a design matrix for statistical analysis using the general linear model on SPM12, allowing each patient to be an independent variable in the model. Age, TIV, and brain dominance for each subject were each entered into the design matrix as a covariate. For this type of study, a more favorable FDR correction for multiple comparisons was employed. Statistical analysis for FA and ADC value comparisons was performed using SPSS software, version 20. Groups were compared using the Pearson χ^2^ test or Student’s t test for normally distributed variables, and with the Mann–Whitney U test for non-normally distributed variables. The results were considered statistically significant if the *p*-value was > 0.05. A Kolmogorov–Smirnov test was used to assess the normality of the distribution of the variables. Brain dominance was used as a covariate for the comparison. Inter-observer reliability for FA and ADC measurements was tested with the Kappa statistics.

## 3. Results

### 3.1. Volumetric Data

Compared with controls, the experimental group showed significantly lower gray matter volume in the mediodorsal nucleus of the left thalamus (*p* < 0.001, corrected for multiple comparisons). Other brain regions showing lower GM values were the left putamen and left insula (*p* < 0.001) (Table 1).

### 3.2. DTI and ADC Values

Inter-observer reliability for FA and ADC measurements showed an interrater agreement of 97% (95% confidence interval (CI) 86–99%) for the thalamus.

Among the 23 students belonging to the control group, three were left-handed and 20 right-handed. The 18 students belonging to the experimental group were all right-handed. FA and ADC values were indexed with the values of the dominant brain side thalamus. Mean indexed thalamic FA values were 1.25 ± 0.14 for the control group and 0.98 ± 0.08 for the study group. Mean indexed thalamic ADC values were 0.91 ± 0.10 for the control group and 1.04 ± 0.12 for the study group. There was a significant difference between the two groups in terms of mean indexed FA and ADC values (*p* < 0.05). Specifically, the mean indexed value of the ADC was higher in the thalamus of the dominant hemisphere of the study group (*p* < 0.05), while the mean indexed values of the FA were lower in the same ROI (*p* < 0.05) (Table 2).

## 4. Discussion

Structural brain changes in response to traumatic events have mainly been reported in persons with psychopathological conditions [24,25], although analogous brain changes have also been found in persons who did not develop psychopathology after trauma exposure [26,27,28]. For instance, Li et al. [27] assessed morphological and functional connectivity changes involving structures related to emotion regulation in earthquake survivors without PTSD. Results revealed in the trauma-exposed group greater grey matter density and altered connectivity than controls, involving prefrontal-limbic regions, particularly the anterior cingulate cortex, the medial prefrontal cortex, the amygdala, and the hippocampus. Accordingly, Ansell et al. [26] demonstrated that increasing cumulative exposure to adverse life events was associated with smaller gray matter volumes in prefrontal and limbic regions, such as the medial prefrontal cortex and the insular cortex putamen and subgenual anterior cingulate regions. Together, these results demonstrated that survivors of traumatic experiences can show specific brain changes of both morphology and functional activity without developing PTSD. Our results are consistent with these findings, showing structural changes of the insula and putamen; importantly, moreover, we also found a significantly lower gray matter volume in the mediodorsal nucleus of the left thalamus compared to controls. Indeed, the structural changes of the thalamus have been previously reported in psychopathological responses to trauma, such as PTSD [29,30] or panic disorders [24]. Consistently, animal studies showed that severe stressful events can determine sensitization in the thalamus, both at the macroscopic and microscopic levels of brain atrophy detection, using MRI in rodents [31]. Studies dealing with the effects of stressful events, such as earthquakes, in non-clinical populations did not reveal volumetric alterations involving the thalamus. Therefore, the present findings could help to shed light on a structure, namely the mediodorsal thalamus, likely playing an important role in brain responses to traumatic experiences that do not necessarily lead to the development of a psychopathological condition.

Several mechanisms have been proposed on the pathophysiology of these thalamic changes, in particular, the activation of the hypothalamic–pituitary–adrenal axis by acute stress. Liu et al. [32] hypothesized that high cortisol levels might affect neurogenesis, leading to a rapid and prolonged decrease in the rate of cell proliferation and an increase in apoptotic cell death (Figure 3).

MRI is the instrumental investigation of choice for the study of the human brain. In addition to standard morphological sequences, which are useful for defining anatomical alterations, there is wide use of advanced sequences that allow both imaging and quantitative information to be obtained about molecular and structural alterations of the brain tissue [33,34,35,36,37,38,39]. Among them, diffusion-weighted and diffusion tensor imaging exploits the signals derived from the movement and directionality of water molecules to evaluate the cellularity and the structural integrity of tissues [40,41,42]. Some researchers have explored the ability of MRI to detect cellular effects of stressful events. Harnett et al. [43] evaluated the DWI white matter changes of the cingulum bundle, uncinate fasciculus, and fornix in a cohort of patients with various medical traumas, finding a correlation between the white matter architecture and post-traumatic stress severity scores. Another group demonstrated significant changes in FA values of the inferior longitudinal fasciculus in PTSD patients compared to trauma-exposed normal controls. Sheik et al. [44] demonstrated that the white matter adjacent to the left thalamus was significantly disrupted in girls with high cortisol activity. Beyond these findings on the white matter, there is still very little scientific understanding of the multimodal MRI evaluation of structural thalamic modifications in response to stress to the best of our knowledge. In our study, we found that the thalamus of the dominant hemisphere showed lower FA values and higher ADC values in earthquake survivors. Relevantly, in persons exposed to stressful experiences leading to PTSD-related symptomatology, MRI volumetric changes of the left thalamus are related to reexperiencing of the traumatic events, with greater reexperiencing being associated with more reduced structural volumes [45]. Moreover, it is also essential to underscore here that the specific involvement of the left thalamus was found in the thalamus–amygdala functional connectivity changes observed in both patients with PTSD and healthy controls receiving oxytocin administration during regulation of emotional responses to presentation of stressful visual stimuli [46]. These results suggest that enhanced connectivity between the thalamus and the amygdala in the left hemisphere can be related to better emotional regulation of responses to visual threats. This seems particularly interesting here, since our sample of earthquake survivors, although showing signs of maladaptive responses such as sleep disorders and anxiety-related responses [11], never developed a psychopathological condition. Indeed, the L’Aquila earthquake survivors’ visual systems showed changes relating to enhanced visual processing of threat-related stimuli (i.e., emotional faces), implying hyperarousal and hypervigilance to threat [13]. Here we could observe structural changes in the left mediodorsal thalamus, a crucial relay station in the processing of relevant, potentially threatening visual stimuli and driving effective emotional regulation responses. On this basis, we might speculate that this intricate pattern of brain changes could account for “brain resilience”, protecting victims from developing clinical symptoms, even if unable to protect them from developing maladaptive responses fully (e.g., sleep disorders and anxiety-related responses), when trauma exposure tends to persist, as in the case of continuous aftershocks following an earthquake.

## 5. Conclusions

In conclusion, our results demonstrate the presence of thalamic structural alterations detected by MRI, confirming the validity of neuroimaging findings as valid imaging biomarkers in the evaluation of neuropsychological response to stress and trauma.

Despite these promising results, questions remain, and there are some limitations to the study that deserve mention. First of all, results should to be interpreted with caution, as they were obtained from a relatively small population, and therefore need to be confirmed also on larger samples. The evaluation of this study was mainly focused on imaging analysis, but it would be interesting to analyze the correlation of the present data with neuropsychological findings as well. Further research should be planned also to investigate possible neuroimaging markers of thalamic plasticity in terms of volume recover after PTSD by extending the neuropsychological and neuroimaging evaluation through a long-term follow-up protocol.

## Figures and Tables

**Figure 1 diagnostics-11-00070-f001:**
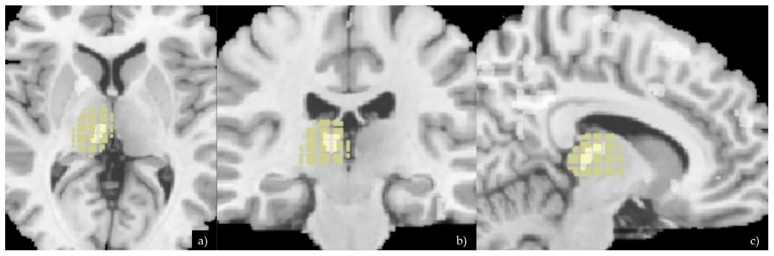
Axial (**a**), coronal (**b**), and sagittal (**c**) planes showing VMB analysis and differences of the thalamus between the two groups. The Xjview anatomy mask specific for the thalamus was then applied on the VBM image in order to highlight the voxels differences in this specific brain region.

**Figure 2 diagnostics-11-00070-f002:**
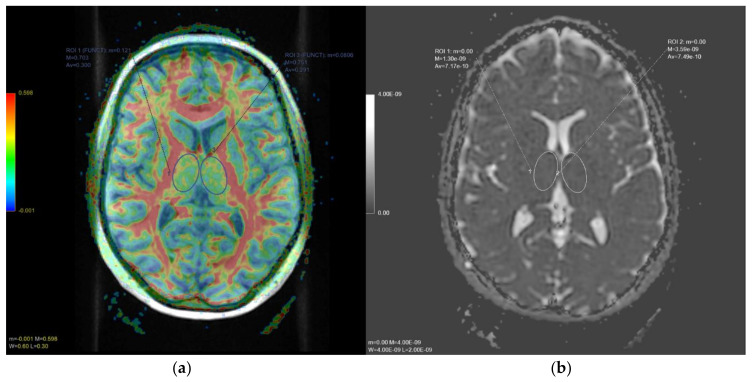
In (**a**), axial diffusion tensor imaging (DTI) slice at thalamic level. The color-coded map represents fractional anisotropy (FA) values (higher FA values in red, depicting white matter tracts, lower FA values in blue). In (**b**), axial apparent diffusion coefficient (ADC) map slice at the same level. Oval regions of interests (ROI) and delineated on both thalami to obtain quantitative FA and ADC values, respectively.

**Figure 3 diagnostics-11-00070-f003:**
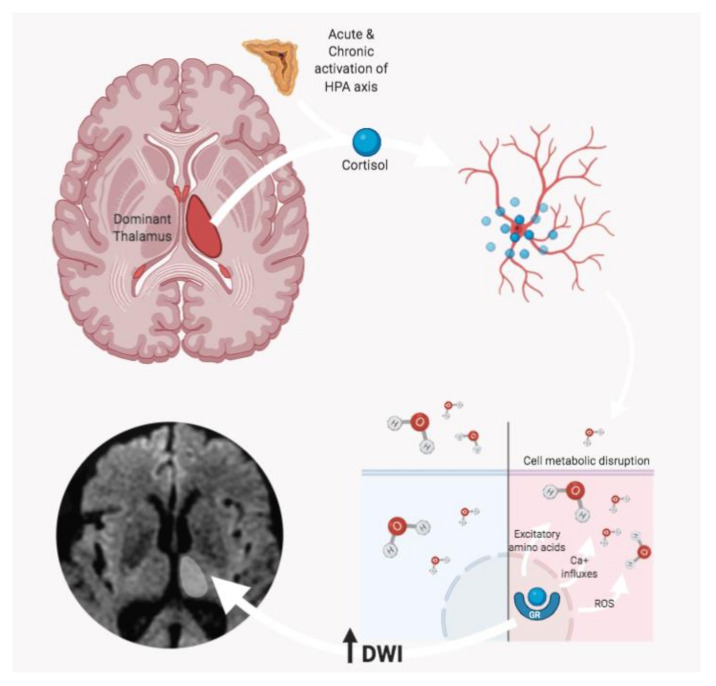
According to Liu et al. [32], stress-related high cortisol levels might affect neurogenesis, leading to a decrease in cell proliferation and an increase in apoptosis rate. In particular, the cascade induced by the glucocorticoid receptor (GR) interaction determines a series of processes including the release of excitatory amino acids, the increase in Ca^+^ influxes, and the release of reactive oxygen species (ROS), which contribute to determining cell metabolic disruption. The consequent cell depletion might be detected as an increase of water diffusivity (due to the lower cellularity), with hyperintense DWI MRI signal.

**Table 1 diagnostics-11-00070-t001:** Monteral Neurological Institute (MNI) coordinates of the anatomical structures with significant VBM differences between the control and study group.

Study > Control Group	x	y	z	Voxel Cluster	*p*-Values
Medio-dorsal Nucleus of the L-Thalamus	−10.5	−21	6		*p* < 0.001
Left Putamen Left Insula	−15 –42	18 15	−6 1.5	694 29	*p* < 0.001 *p* < 0.04

**Table 2 diagnostics-11-00070-t002:** Main differences between the control and study group in terms of ADC and FA.

	ADC Dominant Side	ADC Non-Dominant Side	Indexed ADC (Dominant/Non-Dominant Hemisphere)	FA Values Dominant Side	FA Values Non-Dominant Side	Indexed FA (Dominant/Non-Dominant Hemisphere)
Control Group	7.33 × 10^−6^ ± 0.40	7.90 × 10^−6^ ± 0.55	0.91 ± 0.10	0.36 ± 0.07	0.32 ± 0.06	1.25 ± 0.14
Study Group	7.55 × 10^−6^ ± 0.66	7.19 × 10^−6^ ± 0.51	1.04 ± 0.12	0.31 ± 0.04	0.32 ± 0.04	0.98 ± 0.08
*p*-value	*p* = 0.11	*p* = 0.40	*p* = 0.04	*p* = 0.10	*p* = 0.8	*p* = 0.001

## Data Availability

The data that support the findings of this study are available upon reasonable request request from the corresponding author.

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
