# Peer review of "Multimodal MRI Assessment of Thalamic Structural Changes in Earthquake Survivors"

_diagnostics, 2021, doi:10.3390/diagnostics11010070_

Round 1

Reviewer 1 Report

Thanks for recommending me as a reviewer. The objective of the this study was to investigate the presence of any structural and volumetric changes of the thalamic structures, evident at MRI imaging, in earthquake witnesses. If the authors complete revisions, the quality of the study will be better.

  1. line 29-30:  The author has well described the theoretical background in the'Introduction' section. However, the first two sentences require a reference.

2. line 69-71: "Forty-one (38%) subjects were included, 18 students belonging to the experimental earthquake68 exposed group with a permanent residence in L'Aquila at the time of the 2009 earthquake (8 males and ten females, mean age 24.5±1.8 years) and 23 students belonging to the control group not living in an earthquake-affected area (14 males and nine females, mean age 23.7±2.1 years). Non-parametric between-group comparisons showed that the two groups did not differ for both sex and age." - The proportions of genders are different between groups, and I can't understand the results that there was no difference between groups in the nonparametric test.

3. line 136: The authors stated that the Kolmogorov-Smirnov test was performed in the study. Are the results of the test presented?

4. line 232-234: "While recent animal studies support this interpretation [6], specific neuroimaging investigations are warranted in humans to identify possible MRI imaging biomarkers, as suggested by our findings." - It's awkward because the last sentence of the discussion section ends with a reference. In the final paragraph of the discussion section, it might be a good idea to present the conclusions of your study.

5. What are the limitations of this study? Authors should add the limitations of this study to the'Discussion' section.

Author Response

  1. line 29-30: The author has well described the theoretical background in the 'Introduction' section. However, the first two sentences require a reference. Thank you for the comment, we added references missing in the first paragraphs.
  2. line 69-71: "Forty-one (38%) subjects were included, 18 students belonging to the experimental earthquake68 exposed group with a permanent residence in L'Aquila at the time of the 2009 earthquake (8 males and ten females, mean age 24.5±1.8 years) and 23 students belonging to the control group not living in an earthquake-affected area (14 males and nine females, mean age 23.7±2.1 years). Non-parametric between-group comparisons showed that the two groups did not differ for both sex and age." - The proportions of genders are different between groups, and I can't understand the results that there was no difference between groups in the nonparametric test. Thank you for spotting this. Actually it was an oversight, the matching is only about age, and we have corrected the text
  3. line 136: The authors stated that the Kolmogorov-Smirnov test was performed in the study. Are the results of the test presented? Kolmogorov-Smirnov test was performed to assess the normality of the distribution of the variables. The results are not presented simply because on our SPSS programme the test is automatically performed and the appropriate subsequent test applied. For example, in our cases, the Pearson χ 2 test or the Student's t test for normally distributed variables, and the Mann-Whitney U test for non-normally distributed variables were applied accordingly after the Kolmogorov-Smirnov test results (p > 0,05 for normally distributed variables, p < 0,05 for non-normally distributed variables).
  4. line 232-234: "While recent animal studies support this interpretation [6], specific neuroimaging investigations are warranted in humans to identify possible MRI imaging biomarkers, as suggested by our findings." - It's awkward because the last sentence of the discussion section ends with a reference. In the final paragraph of the discussion section, it might be a good idea to present the conclusions of your study. Thank you for the comment; the last sentence was rewritten to draw the conclusions of the study
  5. What are the limitations of this study? Authors should add the limitations of this study to the'Discussion' section. Thank you for the precious suggestion. We discussed the research limitations

Reviewer 2 Report

The authors reported thalamic structural changes in survivors of 2009 L’Aquila earthquake by using Magnetic resonance imaging (MRI) with multiple analysis technics, such as Voxel-based morphometry (VBM), Diffusion tensor imaging (DTI), Apparent diffusion coefficient (ADC) and Fractional anisotropy (FA). The results of the study are straight forward and consistent with other reported papers. However, the lack of detailed description about the results and figure legends reduces the value of this paper. The following minor comments are for making this manuscript more comprehensive.

Minor

  1. Line 16. Since MRI is an abbreviation of Magnetic Resonance Imaging, “MRI imaging” is a redundant description. Please change it as MRI or Magnetic resonance imaging.
  2. Line 67. What 38% means? Please describe the details how the authors recruited the participants in this study.
  3. Line 67. Was the brain imaging recorded at the same period in all participants? If not, please analyze the relationship between the thalamus size and the time span after the earthquake.
  4. Line 67. 2.2. Participants. Please describe whether these was a significant difference in ethnic between the experimental earthquake-exposed group and control group or not.
  5. Line 98. Fig. 2. Please show representative VBM images in both experimental earthquake-exposed and control brains.
  6. Line 104. Fig. 2 legend. There is a typo. Please change “VMB” to “VBM” because it is an abbreviation of Voxel-based morphometry.
  7. Line 104. Fig. 2. Legend is poor. Please describe details of the VBM data.
  8. Line 104, 127, and 192. Why the order of figures is Fig. 2, Fig. 3, and Fig. 1? Please rename figure numbers according to the order they appear in the paper.
  9. Line 119. Fig. 3. Please show representative DTI and ADC maps in both experimental earthquake-exposed and control brains.
  10. Line 127. Fig. 3. Legend is poor. Please describe details of the DTI and ADC maps. What each color means?
  11. Line 138. Why statistical significance on the FA and ADC map is defined as 0.0083 for the analyses of the thalami? Please describe the rational of this strange threshold or use a common threshold.
  12. Line 160. Table 2. Please describe the p values comparing between control and study groups in each variable, such as ADC dominant side, FA values dominant side, etc.
  13. Line 189. Please add the corresponding paper (Lui et al.) in the References.
  14. Line 193. Fig. 1 legend does not describe enough about the figure. Please appropriately rewrite it.
  15. Line 193. There is a typo. Please change “Acoordi toLui et al.” to “According to Lui et al.” Please add the corresponding paper (Lui et al.) in the References.
  16. Line 202. Does this sentence really describe about Fig.1?
  17. Line 203. Please add the corresponding paper (Harnett et al.) in the References.
  18. Please discuss about the plasticity of the thalamus volume after PTSD. Is the thalamus volume recover after a long span from the traumatic event, such as the earthquake?
  19. Line 275, The article information is not enough. Please revise it with PLoS ONE 2018, 13 (12), e0208152.

Author Response

Line 16. Since MRI is an abbreviation of Magnetic Resonance Imaging, “MRI imaging” is a redundant description. Please change it as MRI or Magnetic resonance imaging. Ok, it was corrected through the text

Line 67. What 38% means? Please describe the details how the authors recruited the participants in this study. Thank you for the suggestion which allows us to better clarify. Out of the initial sample of 107 patients of our previous pilot study, who underwent a complete neuropsychological assessment, the final population of the current study includes 41 patients undergoing instrumental brain MRI examination,

Line 67. Was the brain imaging recorded at the same period in all participants? If not, please analyze the relationship between the thalamus size and the time span after the earthquake. Thanks for the clarification requested. Yes, all patients underwent an MRI examination during a recruitment period of approximately three months, so all the tests were acquired eight years after the traumatic event.

Line 67. 2.2. Participants. Please describe whether these was a significant difference in ethnic between the experimental earthquake-exposed group and control group or not. Thank you for pointing this out. All participats in both the study and control group were Caucasian Italian students, and we specified it in the M&M section

Line 98. Fig. 2. Please show representative VBM images in both experimental earthquake-exposed and control brains. Voxel-based morphometry is a computational approach to neuroanatomy that measures differences in local concentrations of brain tissue, through a voxel-wise comparison of multiple brain images. In our study, we compared the VBM images of these two groups, so the final image is the result of the comparisons between these two. In addition, we also applied a mask that included the voxel-differences in the thalamus and no other brain regions.

Line 104. Fig. 2 legend. There is a typo. Please change “VMB” to “VBM” because it is an abbreviation of Voxel-based morphometry. Ok, it was corrected

Line 104. Fig. 2. Legend is poor. Please describe details of the VBM data. Thank you. We added more details of the figure legend to make it clearer (highlighted in the text).

Line 104, 127, and 192. Why the order of figures is Fig. 2, Fig. 3, and Fig. 1? Please rename figure numbers according to the order they appear in the paper. Ok, it was corrected

Line 119. Fig. 3. Please show representative DTI and ADC maps in both experimental earthquake-exposed and control brains. Actually, the results of our data show significant differences on the quantitative values extrapolated from the ROIs after postprocessing, so a mere qualitative evaluation would not highlight clear imaging differences between the two groups. However, if the reviewers deem it appropriate, we will be happy to include a comparative imaging assessment as well.

Line 127. Fig. 3. Legend is poor. Please describe details of the DTI and ADC maps. What each color means? Thank you for the suggestion, the figure legend was described more in details

Line 138. Why statistical significance on the FA and ADC map is defined as 0.0083 for the analyses of the thalami? Please describe the rational of this strange threshold or use a common threshold. Thank you for your comment. Actually, 0.0083 was the threshold applied for the ADC and FA maps analysis (processing threshold), while for the statistical significance the threshold was the usual p value=0.05. We have rewritten the relevant passages in “image analysis” and “statystical analysis” sections to clarify the error

Line 160. Table 2. Please describe the p values comparing between control and study groups in each variable, such as ADC dominant side, FA values dominant side, etc. Thank you for your revision. We provided all the values as requested (highlighted in the text)

Line 189. Please add the corresponding paper (Liu et al.) in the References. Ok, added

Line 193. Fig. 1 legend does not describe enough about the figure. Please appropriately rewrite it. Thank you for your suggestion, we developed the legend further

Line 193. There is a typo. Please change “Acoordi to Liu et al.” to “According to Liu et al.” Please add the corresponding paper (Liu et al.) in the References. Ok, done

Line 202. Does this sentence really describe about Fig.1? Thank you for this annotation. The reference to Fig. 1 was inserted by mistake and removed

Line 203. Please add the corresponding paper (Harnett et al.) in the References. Thank you for noting, it was added

Please discuss about the plasticity of the thalamus volume after PTSD. Is the thalamus volume recover after a long span from the traumatic event, such as the earthquake? Thanks for the interesting question. Indeed, in the light of our results, this is undoubtedly one of the aspects that need further evaluation, and that currently does not find significant literature evidence. We therefore look forward to future studies through a longitudinal control of the patients in the cohort. We have highlighted this limitation in the discussion.

Line 275, The article information is not enough. Please revise it with PLoS ONE 2018, 13 (12), e0208152. Ok, corrected

Round 2

Reviewer 1 Report

Thanks for recommending me as a reviewer. The objective of the this study was to investigate the presence of any structural and volumetric changes of the thalamic structures, evident at MRI imaging, in earthquake witnesses. The authors have faithfully completed the revision by reflecting the suggestions.